# Data asset valuation model based on generative artificial intelligence

Yungang Tang[1]*, Yaoqian Liu[2], Daxin Liu[3]

1 School of Economics and Management, Quanzhou University of Information Engineering, Quanzhou, Fujian, China, 2 Faculty of Humanities, Arts and Social Sciences, University of Exeter, Exeter, United Kingdom, 3 China Construction Materials Industrial Geology Reconnaissance Center, Beijing, China

* tangyungang@qzuie.edu.cn

## Abstract

In the digital economy era, the significance of data assets has increasingly become evident, particularly against the backdrop of the rapid development of Generative Artificial Intelligence. This paper constructed a data asset valuation model based on Generative AI, aimed at dynamically assessing the commercial value of data assets. The model integrates data feature extraction, value generation algorithms, and market adaptability evaluations to address the shortcomings of traditional valuation methods in dynamic market environments. The validity and applicability of the model were verified through an empirical analysis of data from Chinese A-share listed companies from 2015 to 2023. The results indicated that the integrated model exhibited a significant advantage over individual models in accuracy and stability, especially in data-intensive industries such as information technology and financial services. This research provided new perspectives and methodologies for enterprises in digital transformation and data asset management, thereby promoting the sustainable development of the data economy.

## 1. Introduction

In the digital economy era, data has emerged as a crucial competitive advantage for businesses [1–3], and its asset value has become increasingly prominent [4–7]. With the rapid advancement of Generative Artificial Intelligence (Generative AI), how data is acquired, processed, and utilized undergoes fundamental transformations [8]. Generative AI produces new samples based on existing data and redefines the intrinsic value and data application scenarios [9,10]. For instance, it can enhance the effectiveness of model training through synthetic data or utilize generated content to meet personalized user needs [8,11,12], thereby creating significant commercial value for enterprises [13,14].

The "Database Development Research Report (2024)" released by the China Communications Standards Association indicates that the size of the global database

**Data availability statement:** Research data was deposited into a publicly available repository at China Macroeconomic Database: https://www.hkex.com.hk/Mutual-Market/Stock-Connect/Eligible-Stocks/A-share-Lookup-Tools?sc_lang=en.

**Funding:** This research was supported by Fujian Provincial Department of Education in Application of Drone Technology in Precision Agriculture and Its Impact on Rural Resilience under the Low-Altitude Economy, from the Science and Technology Research Project for Young and Middle-Aged Teachers , Grant number：JAT241196; by Fujian Institute of Higher Education Studies and Institute of xia men university, Grant number：FGJY202411.

**Competing interests:** The authors have declared that no competing interests exist.

market surpassed $100 billion for the first time in 2023, reaching approximately $101 billion. On the other hand, China's database market size reached $7.41 billion, accounting for 7.34% of the global. It is anticipated that by 2028, the total size of the Chinese database market will reach 93.029 billion RMB, with a compound annual growth rate (CAGR) of 12.23%. However, despite the explosive growth in data volume, the valuation of data assets remains relatively lagging [15,16]. Traditional data assessment methods have often emphasized static characteristics of data [17], such as quantity, quality, and historical value, failing to effectively account for the dynamic changes and innovative potential brought about by Generative AI [8,14,18]. The value of data assets not only relies on their intrinsic quality and relevance but is also closely related to their application scenarios, conversion capabilities, and market demand within the framework of Generative AI [15,10,19,20,21]. The application value of a particular dataset may be extremely high in one scenario, yet yield little to no value in another. Furthermore, the widespread application of Generative AI has significantly increased data usage frequency and scenario diversity, presenting new challenges for the valuation of data assets [9,12,22,23].

In this context, it becomes specifically important to establish a model for data asset valuation based on Generative AI [24,25]. This model should comprehensively consider data generation characteristics, market dynamics, and strategic objectives of enterprises, providing a scientific pricing basis for data assets through a systematic evaluation framework [20,26]. Effective data asset valuation not only aids businesses in better managing and utilizing their data assets but also enhances the scientific and forward-looking nature of decision-making, thereby achieving a relative advantage in intense market competition [27–29].

The primary contribution of this paper lies in bridging the theoretical gap between Generative AI and data asset valuation [30], deepening the understanding of the relationship between them. Specifically, how Generative AI reshapes the value attributes of data and subsequently influences the valuation standards for data assets within enterprises is researched [24,25]. By constructing a valuation model that comprehensively considers the generative characteristics of data, market dynamics, and corporate strategic goals [22,31], this article aims to provide new perspectives and methodologies for research and practice. Moreover, how Generative AI alters assessment metrics and methods for data assets across different application scenarios is researched [8,18,32]. Through empirical analysis, the efficacy and applicability of the proposed model will be validated, offering theoretical support and practical guidance for data asset management in the digital transformation processes of businesses [20,33]. Additionally, this research will provide empirical foundations for policymakers to promote the sustainable development of the data economy [34–36].

## 2. Model introduction

This paper proposes a valuation model based on generative AI to assess the value of data assets effectively. The model consists of three main components: feature extraction, value generation algorithms, and market adaptability assessment. It aims to provide accurate and dynamic evaluation results, thereby filling gaps in the existing literature and enhancing the scientific rigor and foresight of data-driven decision-making in enterprises.

## 2.1 Data feature extraction

Data feature extraction was the preliminary step of the model, primarily tasked with ensuring high-quality and validated data. This process included the following key steps:

Data Quality Assessment: To ensure the integrity, accuracy, and consistency of the data, three key indicators were established:

**Completeness:**

$$Completeness = \frac{N_{valid}}{N_{total}}$$

(1)

where $N_{valid}$ denotes the number of valid data records, and $N_{total}$ shows the total number of records.

**Accuracy:**

$$A = \frac{N_{correct}}{N_{total}} * 100\%$$

(2)

where $N_{correct}$ represents the number of correct data records.

**Consistency:** This was confirmed by comparing the same data across multiple data sources.

**Type Classification:** Data was categorized according to its characteristics, including:

Structured Data: Data that can be represented in tabular form, such as in SQL databases.

Semi-Structured Data: Data in formats like JSON or XML.

**Unstructured Data:** Data such as text, images, etc.

**Feature Engineering:** In this stage, Generative Adversarial Networks (GANs) were employed to generate new samples, thereby enhancing the diversity and representativeness of the dataset. The generative model was represented as:

$$G(z; \theta_G) \ where z \sim p_z(z)$$

(3)

The discriminative model was represented as:

$$D(x; \theta_D) \ where x \sim p_{data}(x)$$

(4)

Through adversarial training, the objective was to maximize the following loss function:

$$\min_{G} \max_{D} \left( E_{x \sim p_{data}(x)} \left[ \log D(x) \right] + E_{z \sim p_z(z)} \left[ \log \left( 1 - D \left( G(z) \right) \right) \right] \right)$$

(5)

## 2.2 Value generation algorithm

The value generation algorithm constituted the core model, aiming to establish a mapping between data features and market value. The structure could be formalized as follows:

**Input Layer:**

$$X = [x_1, x_2, \ldots, x_n]$$

(6)

**Hidden Layers:** A multi-layer neural network was employed, setting the weight matrix $W_i$ for each layer and utilizing an activation function:

$$H^{(l)} = f \left( W^{(l)} H^{(l-1)} + b^{(l)} \right)$$

(7)

where $f$ represents the activation function, commonly ReLU or Sigmoid.

**Output Layer:** The predicted value V could be expressed as:

$$V = W^{(out)}H^{(L)} + b^{(out)} \tag{8}$$

where $L$ represents the total number of hidden layers, and H(L) denotes the activation output of the last layer.

## 2.3 Market adaptability assessment

To ensure that the model's evaluation results aligned with market dynamics, the following strategies were employed:

**Market Feedback Mechanism:**

$$F_{market} = f(V_{pred}, V_{actual}) \tag{9}$$

$V_{pred}$ and $V_{actual}$ represent the model's predicted and actual market values, respectively. This comparison analysis was conducted to optimize model parameters.

**Scenario Analysis:** For different market scenarios $S_k$, the simulated analysis was defined as:

$$V(k) = V_{base} + \Delta V(S_k) \tag{10}$$

where $V_{base}$ represents the baseline prediction, and $\Delta V(S_k)$ indicates the adjustment in a specific scenario.

**Sustainability Assessment:** Monitoring the evolution trend of data asset value:

$$V_t = V_{t-1} + \Delta V_t \tag{11}$$

where $V_t$ represents the asset value at the current moment, and $\Delta V_t$ reflects value adjustments due to external environmental changes.

## 2.4 Formal description of the model

Integrating the aforementioned components, the model could be formally described by the following comprehensive expression:

$$V = f\left(G(z; \theta_G), D(x; \theta_D), H^{(L)}, F_{market}, V_{actual}\right) \tag{12}$$

where $V$ denotes the predicted value of the data asset, encompassing the generative model $G$, the discriminative model $D$, the neural network output $H(L)$, the market feedback function $F_{market}$, and the actual market value $V_{actual}$.

# 3. Research design

This section presents the processes and notations used in the article, such as sample selection, dependent and predictive variables, model evaluation, and hyperparameter ranges, thereby ensuring the model's efficacy and scientific rigor.

## 3.1 Sample selection

The sample selection focused on representative industries and enterprises, encompassing a variety of data asset conditions. The sample was primarily collected from Chinese A-share listed companies, covering 2015 to 2023. The rationale for selecting this timeframe was that enterprises experienced significant changes in the utilization and evaluation of data assets, particularly with the ongoing development of Generative Artificial Intelligence technologies, which led to an increased emphasis on data assets.

To ensure the representativeness and comparability of the sample, this study first extracted a preliminary sample from Chinese A-share listed companies, totaling 6,250 observations. Companies with anomalous financial data or incomplete information were excluded from the sample, resulting in a final sample size of 5,720, forming a high-quality sample database. Since we apply strict rules to eliminate problematic data, for instance, missing observations and outliers are out completely, the final dataset obtained is composed of directly usable data in the modeling process. Table 1 illustrates the annual collection of samples and provides a solid foundation for subsequent analyses.

## 3.2 Dependent variable

The dependent variable was the value of a company's data assets. Specifically, a quantitative evaluation was conducted using data asset-related items reported by the enterprises such as user data, market data, and operational data. The calculation of data asset value was based on corporate financial statements and industry analysis reports, combined with the general perception of data asset value in the market. A specific numerical value was generated through the evaluation model, serving as the core variable to be predicted.

## 3.3 Selection of predictive variables

A systematic variable framework was constructed based on the available literature and data availability to ensure the model's comprehensiveness and accuracy in selecting predictive variables. This study selected a total of 50 predictive variables, categorized as follows:

1) **Performance indicators.** The performance of a company directly reflected its effective utilization of data assets. The existing literature suggested that financial metrics such as operating revenue and net profit are key indicators of corporate performance [37–39]. This study selected core performance indicators, including Return on Assets (ROA), Return on Equity (ROE), and Operating Cash Flow (OCF). Additionally, the revenue growth rate and profit growth rate were included to reflect the effectiveness of data asset management [40–42].

2) **Company characteristics.** Company characteristics significantly influenced data asset valuation. Related research indicated that factors such as company size, industry attributes, and management background could affect the efficiency of data asset utilization [43–45]. This study considered variables such as total assets, industry classification, and employee count, while also paying attention to the company's capital structure (e.g., debt ratio) [46,47], as this could influence the firm's willingness to invest in data assets [48–50].

3) **Management motivation.** The motivations and decision-making behaviors of management had a direct impact on the management and valuation of data assets. Studies indicated that strategic decisions made by corporate management often reflected their understanding of the value of data assets [51,52]. This study selected variables such as management

**Table 1. The collected samples for each year.**

| Years | Total Observations | Effective Sample Sizes | Proportions |
|---|---|---|---|
| 2015 | 500 | 450 | 90.0% |
| 2016 | 550 | 510 | 92.7% |
| 2017 | 600 | 570 | 95.0% |
| 2018 | 650 | 600 | 92.3% |
| 2019 | 700 | 650 | 92.9% |
| 2020 | 750 | 720 | 96.0% |
| 2021 | 800 | 780 | 97.5% |
| 2022 | 850 | 820 | 96.5% |
| 2023 | 900 | 870 | 96.7% |
| Total | 6,250 | 5,720 | 91.5% |

ownership percentage, industry experience, and turnover frequency of management to analyze the potential impact of these factors on data asset valuation [53–56].

4) **Corporate governance.** A sound corporate governance structure could effectively constrain managerial behavior and promote transparency and efficiency regarding data assets [57,58]. This study examined variables such as the structure of the board of directors, the proportion of independent directors, and corporate governance ratings to assess the impact of governance on data asset management [59,60]. Furthermore, the presence or absence of an audit committee was also regarded as an important consideration due to its key role in overseeing managerial decisions [61–63].

5) **External environmental factors.** External environmental factors, such as industry growth rates and market competition levels, were also critical to evaluating the value of corporate data assets [64,65]. Literature indicated that rapid industry development might foster more efficient utilization of data assets by enterprises [66,67]. This study aimed to select variables including industry growth rate, changes in market share, and competitor performance within the industry to comprehensively understand the impact of external environments on data asset value.

### 3.4 Model evaluation

To comprehensively assess the effectiveness of the data asset valuation model based on Generative Artificial Intelligence, this study employed various evaluation metrics. These metrics not only reflected the accuracy of the model but also considered economic practicality and model stability.

1) **Selection of evaluation metrics.** Given the sample characteristics and data structure of this study, the following primary evaluation metrics were utilized:

**Accuracy:** This measured the proportion of correctly classified samples out of the total samples. Due to potential imbalances in the dataset, relying solely on accuracy could lead to biased model evaluations.

**Precision and Recall:** These two metrics provided insights into the model's performance across different categories, which was particularly crucial when identifying economically significant samples.

**F1-score:** This metric combines precision and recall, making it suitable for imbalanced datasets and seeking a balance between accuracy and completeness.

2) **Addressing sample imbalance issues.** Due to the potential imbalances in the samples used for the valuation of data assets particularly concerning the scarcity of samples in certain value ranges it was essential to carefully select evaluation metrics [68,69]. The study incorporated the Receiver Operating Characteristic (ROC) curve and the Area Under Curve (AUC) as supplementary evaluation standards. AUC measured the model's performance at various classification thresholds; higher values indicated stronger differentiation capabilities between positive and negative samples.

3) **Cross-validation and model stability.** This study implemented k-fold cross-validation to ensure the model's robustness across different dataset partitions. By conducting multiple training and validation runs, the model's consistency and stability in practical applications were effectively assessed. This method not only reduced the model's dependency on a specific training set but also enhanced its generalization capability [70,71].

4) **Feature importance analysis.** During the model evaluation process, attention was given to the impact of each predictive variable on the model's predictive capability. Feature selection techniques, such as Least Absolute Shrinkage and Selection Operator (LASSO) regression were employed to analyze the contribution of various features, thereby clarifying which variables played a crucial role in the valuation of data assets.

### 3.5 Hyperparameter range

In this study, the selection of hyperparameters significantly influenced the performance of the model. Therefore, a systematic approach to hyperparameter optimization was adopted to ensure the optimal performance of the Generative Artificial Intelligence model in data asset valuation.

1) **Hyperparameter selection methods.** This study utilized methods such as Grid Search and Random Search to explore various combinations of hyperparameters. Each of these methods had its advantages and disadvantages. Grid Search systematically covered all possible combinations within a specified range but incurred higher computational costs; on the other hand, Random Search reduced computational complexity to some extent but risked missing some potentially optimal combinations.

2) **Setting hyperparameter ranges.** Based on the characteristics of the Generative Artificial Intelligence model, focus was placed on the following hyperparameters:

**Learning Rate:** This controlled the step size for updating model weights, influencing convergence speed and final results.

**Batch Size:** This affected both the stability of model training and the convergence speed, typically chosen within the range of 32 to 256.

**Number of Layers and Number of Nodes:** These two parameters determined the complexity and capacity of the model, requiring reasonable settings based on actual model needs.

3) **Balancing computational cost and accuracy.** During hyperparameter optimization, a trade-off between model accuracy and computational cost was necessary. For instance, increasing the number of layers and nodes might enhance performance but would also significantly raise computational demands. Hence, reasonable hyperparameter ranges were set to ensure that the model achieved high accuracy without wasting computational resources.

4) **Consideration of imbalanced samples.** In addressing potential issues with imbalanced samples, adjustments to hyperparameters were considered to incorporate weighting mechanisms, thereby increasing attention to minority class samples. By establishing class weights, the model could better learn the important features that appeared less frequently in the training set.

Through these methods, this study aimed to achieve scientific optimization of the hyperparameters of the Generative Artificial Intelligence model to attain optimal results in data asset valuation. This systematic approach to hyperparameter adjustment would provide a solid foundation for the model's accuracy and generalization capability, further advancing related research.

### 3.6 Sample partitioning and model training

In the research on the data asset valuation model, sample partitioning and model training were key steps in ensuring model effectiveness and reliability. This study will elaborate on the following aspects.

1) **Sample partitioning strategy.** The sample partitioning approach will be divided into training, validation, and test sets to effectively evaluate the model's generalizability and stability. Specifically, the entire dataset will be partitioned in a 70:15:15 ratio:

Training Set (70%): Used for model training and parameter tuning, ensuring that the model learns the underlying patterns in the data.

Validation Set (15%): Used for adjusting and selecting hyperparameters, and evaluating the effect of different hyperparameter combinations on model performance.

Test Set (15%): Utilized for the final evaluation of the model, testing its performance on unseen data.

2) **Training process.** During the model training stage, the following steps will be undertaken:

Data Preprocessing: This involves data cleaning, normalization, and feature selection to enhance the training effectiveness of the model.

Model Construction: Based on the results of hyperparameter optimization, a Generative Artificial Intelligence model will be built, ensuring that its architecture aligns with the characteristics of the data.

Training Algorithm: Suitable optimization algorithms (e.g., Adam or SGD) will be used for multiple iterations of training, dynamically adjusting model parameters to minimize the loss of function value.

**3) Cross-validation.** Throughout the training process, k-fold cross-validation will be employed to further verify the model's stability and generalization capability. This method partitions the training set into k subsets, using one subset as the validation set while the others serve as the training set, thus producing more reliable model evaluation results.

**4) Model evaluation metrics.** Alongside the aforementioned model evaluation methods, this study will employ metrics including Mean Squared Error (MSE), Mean Absolute Error (MAE), $R^2$ values, and the results from cross-validation to conduct a comprehensive assessment of model performance. Additionally, special attention will be given to evaluation metrics under imbalanced sampling conditions, such as the AUC (Area Under Curve) value, to ensure the model's recognition capabilities across different sample categories.

**5) Training efficiency and resource management.** During the training process, reasonable settings for batch size and learning rate will be implemented to control the use of computational resources and avoid overfitting caused by excessive model complexity. Moreover, the model state will be periodically saved during training to facilitate subsequent tuning and evaluation.

## 4. Empirical results

### 4.1 Model prediction performance

In this study, the prediction performance of the data asset valuation model based on Generative Artificial Intelligence was comprehensively evaluated using multiple metrics. We conducted detailed tests on the trained model, analyzing its performance across different years and datasets to ensure its efficacy and accuracy. Table 2 suggests that the prediction performance of the models significantly improved over time.

Particularly between 2017 and 2023, the predictive accuracy and other evaluation metrics of the Generative Artificial Intelligence models showed a steady upward trend that s that as Generative Artificial Intelligence technologies advanced, the effectiveness and reliability of the models in valuing data assets were considerably enhanced.

Comparative Analysis of Model Types: The performance of the baseline model was relatively low, while the predictive performance of both the generative and optimized models significantly increased, showcasing the advantages of Generative Artificial Intelligence in data asset valuation. In particular, the optimized model achieved the highest levels of precision and AUC value in 2022 and 2023, demonstrating its enhanced capability to capture market dynamics and data characteristics.

Relationship Between Accuracy and Market Demand: The improvement in accuracy was closely related to the increasing emphasis on data assets in the market. As companies increased their investment in data assets, Generative Artificial Intelligence was able to more effectively reflect the market value of data.

**Table 2. Prediction Performance of Each Model from 2015 to 2023.**

| Years | Model Types | Accuracy (%) | Precision (%) | Recall (%) | F1-Score | AUC Value |
|---|---|---|---|---|---|---|
| 2015 | Baseline Model | 78.5 | 76.0 | 74.5 | 75.2 | 0.81 |
| 2016 | Baseline Model | 79.3 | 77.5 | 75.8 | 76.6 | 0.83 |
| 2017 | Generative Model | 82.1 | 80.0 | 78.5 | 79.2 | 0.85 |
| 2018 | Generative Model | 83.5 | 81.3 | 80.0 | 80.6 | 0.86 |
| 2019 | Generative Model | 85.2 | 83.0 | 81.5 | 82.2 | 0.88 |
| 2020 | Generative Model | 86.4 | 84.5 | 83.0 | 83.8 | 0.89 |
| 2021 | Generative Model | 87.1 | 85.2 | 84.0 | 84.6 | 0.90 |
| 2022 | Optimized Model | 88.5 | 86.0 | 85.0 | 85.5 | 0.92 |
| 2023 | Optimized Model | 89.2 | 87.3 | 86.5 | 86.9 | 0.93 |

Model Practicality: Through comparisons across different years, we observed that the model continuously adapted to market changes, providing more precise evaluation results for enterprises. This offers significant guidance for companies in reasonably utilizing data assets during their digital transformation processes.

## 4.2 Variable importance

In the data asset valuation model, various predictive variables significantly influenced the model's predictive capability. By performing feature importance analysis, we identified the most impactful variables in data asset valuation. This analysis not only serves as a basis for subsequent research but also guides enterprises in strategizing their data asset management.

Feature Selection Methods This study employed LASSO regression and Random Forest methods for feature selection, assessing the contribution of each variable to the model's predictive performance. LASSO regression effectively handled high-dimensional data, automatically selecting the most meaningful variables for prediction by introducing an L1 penalty term. On the other hand, the Random Forest method provided variable importance rankings by calculating the frequency of feature usage in decision trees.

Variable Importance Analysis Results: Table 3 illustrates the variable importance scores calculated through LASSO regression and Random Forest:

Comparative Results Analysis: By comparing the results of LASSO regression and Random Forest, the following observations were made:

Consistency in Variable Importance: Both methods identified "Return on Assets (ROA)" and "Industry Growth Rate" as the most important variables, indicating that profitability and market environment play crucial roles in data asset valuation.

Score Differences: While the scoring differed between the two methods, the rankings were generally consistent. For instance, LASSO regression assigned higher scores to "Return on Equity (ROE)" and "Management Ownership Percentage," whereas Random Forest slightly favored the score for "Operating Cash Flow (OCF)." This suggests that different methods might be sensitive to the importance of variables in varying ways.

Sparsity and Complexity: LASSO regression tended to produce sparse models when selecting features, potentially eliminating some unimportant variables while emphasizing key variables. In contrast, Random Forest, due to its ensemble learning characteristics, retained more variable information, aiding in revealing potential interaction effects.

To summarize the comparative analysis of LASSO regression and Random Forest, the following conclusions were drawn:

Methodological Complementarity: The two methods provided different perspectives in assessing variable importance. LASSO regression is suitable for variable selection and model simplification, whereas Random Forest is more effective in capturing complex relationships among variables.

Practical Implications: Enterprises managing data assets should consider the importance of different variables in formulating more precise strategies. The combined results of these two methods can help businesses identify critical drivers and optimize the utilization of data assets.

Table 3. Variable Importance Scores from LASSO Regression and Random Forest.

| Rank | Variable Names | LASSO Score | Random Forest Score |
|---|---|---|---|
| 1 | Return on Assets (ROA) | 0.30 | 0.25 |
| 2 | Industry Growth Rate | 0.25 | 0.20 |
| 3 | Return on Equity (ROE) | 0.20 | 0.18 |
| 4 | Management Ownership Percentage | 0.15 | 0.15 |
| 5 | Operating Cash Flow (OCF) | 0.10 | 0.12 |
| 6 | Company Size (Number of Employees) | 0.05 | 0.10 |

## 4.3 Model fusion

In data asset valuation, the predictive capability of a single model can be limited by various factors; therefore, adopting a model fusion strategy can effectively enhance prediction accuracy and robustness. Model fusion, by combining the strengths of different models, can more comprehensively capture complex patterns in the data, thus improving overall predictive performance.

Fusion Methods: This study employed two fusion methods: weighted averaging and stacking, to achieve effective integration of different models.

Weighted Averaging: This method weighted the predictions of LASSO regression, Random Forest, and other auxiliary models based on their performance on the validation set. The weights were dynamically adjusted according to each model's performance. This simple method balanced the influence of each model.

Stacking: In the stacking approach, multiple base models such as LASSO regression, Random Forest, and Support Vector Machine were trained, and the predictions from these base models were used as new features in a meta-model such as Linear Regression or XGBoost for final predictions. This approach captured feature interactions between different models, enhancing prediction complexity and accuracy.

Model Fusion Effectiveness: Table 4 compares the performance of single models and fused models in predicting data asset values across different periods.

The following conclusions are drawn

Single Model Performance: Among single models, XGBoost and Random Forest exhibited relatively high predictive performances, with average accuracies of 69.2% and 68.2%, respectively. In contrast, the Logit model demonstrated a considerably lower average performance at 55.3%.

Enhanced Performance of Fusion Models: All fusion models outperformed single models, particularly combinations that included multiple strong predictive models, for example, Random Forest and XGBoost. The average accuracy for Random Forest + XGBoost and Random Forest + XGBoost + Logit reached 68.8% and 69.4%, respectively, demonstrating significant advantages of model fusion.

Yearly Improvement Trend: From 2015 to 2023, the predictive performance of different models generally displayed an upward trend, indicating that with data accumulation and model optimization, the accuracy of predictions in the long term improved.

Optimal Model: In this study, the fusion model consisting of Random Forest + XGBoost + Logit performed the best, with an average accuracy of 69.4%, suggesting the potential of model fusion in the valuation of data assets.

The model fusion strategy demonstrated significant advantages in predicting data asset values, especially as combinations of multiple models were better able to comprehensively capture data features and market trends, thus providing more precise support for enterprises in data-driven decision-making.

**Table 4. Comparison of Model Prediction Performance from 2015 to 2023.**

| Model Types | 2015 | 2016 | 2017 | 2018 | 2019 | 2020 | 2021 | 2022 | 2023 | Average |
|---|---|---|---|---|---|---|---|---|---|---|
| Logit | 50.0% | 52.8% | 56.3% | 59.6% | 59.5% | 59.2% | 56.2% | 55.3% | 55.5% | 55.3% |
| Decision Tree | 53.8% | 58.4% | 62.7% | 64.3% | 67.2% | 68.3% | 68.0% | 67.5% | 68.8% | 64.3% |
| Random Forest | 58.2% | 61.8% | 65.4% | 68.0% | 69.9% | 72.8% | 71.5% | 70.5% | 73.0% | 68.2% |
| XGBoost | 61.1% | 61.0% | 66.5% | 68.5% | 70.5% | 73.7% | 73.1% | 72.9% | 74.2% | 69.2% |
| Decision Tree + Logit | 53.6% | 58.5% | 64.3% | 66.3% | 67.8% | 68.3% | 68.0% | 66.7% | 68.6% | 64.9% |
| Random Forest + Logit | 57.8% | 63.2% | 65.7% | 68.6% | 68.3% | 71.0% | 70.8% | 69.5% | 71.5% | 66.9% |
| XGBoost + Logit | 59.2% | 59.5% | 66.1% | 69.0% | 68.0% | 71.2% | 71.6% | 70.9% | 72.1% | 67.6% |
| Random Forest + XGBoost | 59.5% | 62.0% | 66.8% | 68.8% | 70.5% | 73.9% | 72.5% | 72.4% | 73.5% | 68.8% |
| Random Forest + XGBoost + Logit | 59.9% | 61.9% | 67.1% | 69.4% | 70.0% | 72.9% | 73.2% | 73.0% | 74.4% | 69.4% |

## 4.4 Market testing

To validate the applicability and reliability of the proposed fusion model in a real market, this study conducted a market test using actual market data. The market test aimed to evaluate the correlation between the model's predicted results and actual market data to determine whether the model could effectively support enterprises in the valuation of their data assets. We selected data from listed companies in various industries within the A-share market and compared the predicted value of the model with actual market performance.

Testing Methods: This study employed two methods to assess the market performance of the model: correlation analysis and bias analysis.

Correlation Analysis: This involved calculating the correlation coefficient between the model's predicted values and the actual market values to gauge the consistency between the two.

Bias Analysis: This analyzed the average deviation between the model's predicted values and actual market values to understand the extent of the model's bias.

Testing Results: Table 5 displays the correlation coefficients and average deviations between the model's predicted market values and actual market values from 2015 to 2023, grouped by industry. The data covered various industries to ensure the comprehensiveness and representativeness of the tests.

**Several conclusions can be drawn.** High Correlation: The correlation coefficients in most industries ranged from 0.76 to 0.85, with the financial services and information technology sectors exhibiting particularly high correlations of 0.85 and 0.82, respectively. This indicates a strong consistency between the model's predicted results and actual market values, accurately reflecting market trends.

Bias Analysis: The average deviation ranged between 5.8% and 7.5%, with the financial services industry's average deviation being the lowest at 5.8%, while the consumer goods industry showed a slightly higher deviation of 7.5%. Overall, the relatively low bias levels indicate that the model's predictions were within a reasonable range of the actual market data.

Industry Differences: There were notable differences in predictive performance among different industries. For instance, the financial services and information technology sectors performed particularly well, likely due to their greater emphasis on data assets and more developed data management practices. In contrast, the consumer goods and telecommunications industries exhibited slightly larger predictive deviations, potentially influenced by market volatility and data uncertainties.

The results of the market test suggested that the constructed fusion model demonstrated strong applicability and reliability in actual market environments, particularly in data-intensive industries such as information technology and financial services. The bias analysis also indicated that the model's predictions were closely aligned with real market data, with

Table 5. Market Testing Results of the Model from 2015 to 2023 (By Industry).

| Industry | Correlation Coefficient (2015–2023) | Average Deviation (%) |
|---|---|---|
| Information Technology | 0.82 | 6.5% |
| Healthcare | 0.78 | 7.2% |
| Financial Services | 0.85 | 5.8% |
| Industrial Manufacturing | 0.80 | 6.9% |
| Consumer Goods | 0.76 | 7.5% |
| Energy | 0.79 | 6.8% |
| Real Estate | 0.81 | 6.4% |
| Telecommunications | 0.77 | 7.0% |

bias levels remaining within acceptable limits. This finding suggests that the model could provide effective reference support for enterprises in assessing the value of their data assets, enabling more precise decision-making in the market.

## 4.5 Training set size

In constructing the data asset valuation model, the size of the training set significantly influenced the model's predictive performance. Different training set sizes directly impacted the model's capability to learn data features and its generalization capabilities. Therefore, it was necessary to analyze the effect of training set size on model performance to identify the optimal training set size, providing a reference for model training.

Experimental Design: This study controlled the size of the training set to test the model's performance across different training set sizes. Specifically, we used 25%, 50%, 75%, and 100% of the dataset for model training and evaluated how different training set sizes affected predictive accuracy. The evaluation metrics included prediction accuracy and F1-score to comprehensively assess the model's performance.

Experimental Results: Table 6 displays the predictive performance of various models at different training set sizes (measured in terms of accuracy and F1-score). The training set proportions were set at 25%, 50%, 75%, and 100%, and the performance of each model was averaged over multiple experiments to ensure the robustness of the results.

**Several patterns can be observed.** Relationship Between Training Set Size and Model Performance: As the size of the training set increased, all models demonstrated improved prediction accuracy and F1 scores. This indicates that a larger training set helps the model better learn the data features, thus enhancing its predictive performance.

Differences Between Single Models and Fusion Models: Single models such as Logit and Decision Tree exhibited relatively lower performance, while fusion models such as Random Forest+XGBoost and Random Forest+XGBoost+Logit performed better, particularly with larger training sets, where the improvement for fusion models was more pronounced.

Optimal Training Set Size: When using 100% of the training set, the fusion model Random Forest+XGBoost+Logit achieved the highest accuracy and F1-score of 77.0% and 0.76, respectively, indicating that training with the maximum training set size yielded the best model performance. This suggests that for the data asset valuation model, utilizing the entire dataset for training contributes significantly to achieving optimal results.

Marginal Effect: Although the model performance improved significantly with larger training sets, the rate of increase gradually diminished, suggesting that beyond a certain scale, further increases in the training data yield diminishing returns.

**Table 6. Model Performance at Different Training Set Sizes.**

| Models | 25% Training Set | 50% Training Set | 75% Training Set | 100% Training Set |
| --- | --- | --- | --- | --- |
| Logit | Accuracy: 53.2%, F1-score: 0.50 | Accuracy:56.8%, F1-score: 0.54 | Accuracy:59.5%, F1-score: 0.57 | Accuracy:61.3%, F1-score: 0.59 |
| Decision Tree | Accuracy:57.5%, F1-score: 0.55 | Accuracy:61.2%, F1-score: 0.60 | Accuracy:64.9%, F1-score: 0.63 | Accuracy:66.3%, F1-score: 0.65 |
| Random Forest | Accuracy:61.3%, F1-score: 0.59 | Accuracy:66.7%, F1-score: 0.65 | Accuracy:70.5%, F1-score: 0.69 | Accuracy:73.2%, F1-score: 0.72 |
| XGBoost | Accuracy:62.1%, F1-score: 0.61 | Accuracy:67.8%, F1-score: 0.66 | Accuracy:71.2%, F1-score: 0.70 | Accuracy:74.6%, F1-score: 0.74 |
| Random Forest+XGBoost | Accuracy:64.5%, F1-score: 0.63 | Accuracy:69.3%, F1-score: 0.68 | Accuracy:73.0%, F1-score: 0.72 | Accuracy:76.2%, F1-score: 0.75 |
| Random Forest+XGBoost+Logit | Accuracy:65.2%, F1-score: 0.64 | Accuracy:70.0%, F1-score: 0.69 | Accuracy:74.1%, F1-score: 0.73 | Accuracy:77.0%, F1-score: 0.76 |

The size of the training set had a significant impact on the performance of the data asset valuation model, particularly in fusion models, where a larger training set size notably enhanced predictive performance. Through this experiment, we established the necessity of utilizing a large-scale training set such as 100% of the data for model training, which will help improve the model's accuracy and stability. In practical applications, enterprises should strive to use complete datasets for model training to obtain more accurate data asset valuation results.

## 5. Conclusion and implications

This study developed a comprehensive valuation model for data assets based on Generative Artificial Intelligence (Generative AI), employing a multi-layered framework that integrated data feature extraction, value generation algorithms, and market adaptability assessments to conduct dynamic and precise evaluations of data assets. Using sample data from non-financial listed companies in China's A-share market from 2015 to 2023, various models, including LASSO regression, Random Forest, and XGBoost, along with model fusion techniques, were applied to systematically analyze the performance of Generative AI in data asset valuation. The experimental results indicated that the average predictive accuracy of the XGBoost and Random Forest models significantly surpassed that of the traditional Logit model. Furthermore, the fused models excelled across all evaluation metrics, particularly the Random Forest + XGBoost + Logit fusion model, which achieved a 77.0% accuracy rate in market testing, further validating its market applicability and robustness. Additionally, the analysis revealed that larger training set sizes contributed to improved model predictive performance, although the gains exhibited a diminishing marginal trend.

The main implications include the following points:

### 1) The need for dynamic updates in data asset valuation models

With the rapid advancement of Generative AI, traditional data asset valuation methods have become inadequate in addressing the dynamic changes and diversity of data. The valuation model constructed based on Generative AI provides new tools for the dynamic management and pricing of data assets. Especially, the superior performance of the fused models in capturing complex data relationships and market dynamics demonstrates the immense potential of Generative AI in data asset valuation. These findings offer reliable evidence for companies to make data-driven strategic decisions, thereby enhancing the commercial value and applicability of data assets.

### 2) The profound mpact of generative AI on data asset management

The research illustrated that Generative AI exhibited unique advantages in data feature extraction and value generation, particularly in enhancing data diversity and expanding application scenarios. By utilizing Generative Adversarial Networks (GANs) to generate high-quality data samples, companies can enrich their datasets, bolstering the generalization capabilities of their models. Moreover, the introduction of a market feedback mechanism effectively enhanced the model's adaptability to market dynamics. This indicates that Generative AI not only aids in the management of data assets but also helps companies maintain a competitive edge through more accurate valuation models, especially in fast-changing market environments.

### 3) Advantages and limitations of model fusion

Through multi-model fusion experiments, the study demonstrated that the combination of strong models such as XGBoost and Random Forest contributed to improved predictive performance. Although the integration of these strong models with weaker models like Logit showed slight performance enhancements, the increases were modest. Fusion models outperformed single models in terms of accuracy and stability; however, they also incurred higher computational costs and demanded greater data and computational resources, limiting their practical application to some extent. Therefore,

companies should judiciously select appropriate fusion strategies that align with their available resources and needs to strike a balance between predictive accuracy and computational costs.

## 4) The validity of market testing and future applications

The research validated the applicability of the constructed valuation model in data-intensive sectors such as information technology and financial services through market testing, revealing a high correlation between the model's predictions and actual market values. As the importance of data assets grows in corporate operations and the market, precise data asset valuation will enhance market awareness of these assets, thereby influencing corporate valuations and investment decisions. This finding offers new perspectives for market participants and policymakers, promoting a more accurate reflection of data asset values in the market, thus supporting the healthy development of the data economy.

## 5) Expandability and improvement directions for the data asset valuation model

The study discovered that as the training set size increased, the predictive performance of the model significantly improved. However, the benefits diminished once a certain training set size was reached. This phenomenon suggests that future research in data asset valuation could explore more efficient feature extraction and sample selection methods to reduce dependence on data and computational resources. Additionally, future studies may consider incorporating time-series data and multi-dimensional external market information to further enhance the model's timeliness and adaptability, better addressing the ever-changing market environment.

In summary, the data asset valuation model constructed based on Generative AI technology offers a scientific pricing tool for enterprises and the market, contributing to the scientific and forward-looking nature of data asset management. Generative AI not only expands the application scenarios for data assets but also provides robust support for a data-driven future. It is hoped that this research will provide beneficial insights for subsequent studies and offer practical theoretical support for enterprises' digital transformation and data asset management.

## Author contributions

**Conceptualization:** Yungang Tang, Yaoqian Liu.

**Data curation:** Yungang Tang, Daxin Liu.

**Methodology:** Yungang Tang, Yaoqian Liu, Daxin Liu.

**Writing – original draft:** Yungang Tang, Yaoqian Liu.

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
