## [Decision Letter · Decision Letter 0]

17 Apr 2025

Dear Dr. Tang,

Thank you for submitting your manuscript to PLOS ONE. After careful consideration, we feel that it has merit but does not fully meet PLOS ONE’s publication criteria as it currently stands. Therefore, we invite you to submit a revised version of the manuscript that addresses the points raised during the review process.

We look forward to receiving your revised manuscript.

Kind regards,

Fu-Sheng Tsai

Academic Editor

PLOS ONE

Journal Requirements:

3. Please note that your Data Availability Statement is currently missing [the repository name and/or the DOI/accession number of each dataset OR a direct link to access each database]. If your manuscript is accepted for publication, you will be asked to provide these details on a very short timeline. We therefore suggest that you provide this information now, though we will not hold up the peer review process if you are unable.

Reviewers' comments:

Reviewer's Responses to Questions

**Comments to the Author**

1. Is the manuscript technically sound, and do the data support the conclusions?

Reviewer #1: Partly

Reviewer #2: Yes

2. Has the statistical analysis been performed appropriately and rigorously?

Reviewer #1: Yes

Reviewer #2: Yes

3. Have the authors made all data underlying the findings in their manuscript fully available?

Reviewer #1: Yes

Reviewer #2: No

4. Is the manuscript presented in an intelligible fashion and written in standard English?

Reviewer #1: Yes

Reviewer #2: Yes

Reviewer #1: This article builds a generative artificial intelligence-based data asset valuation model based on existing literature, which integrates data feature extraction, generative value generation algorithm and market adaptive valuation to dynamically assess the business value of data assets. The conclusions of the article show that the model can address the shortcomings of traditional valuation methods in data asset valuation. Overall, the article is well written, the analysis is solid and the conclusions are challenging and bring a lot of new information to the field.

However, there are certain issues, such as the sample and analysis methods of the study, that deserve in-depth discussion. There are four issues that require further research.

1.What is the basis for the composition of the three dimensions of the model? What are the criteria for dimension selection?Please elaborate.Please elaborate.

2.Do metrics such as user data, market data, and operational data provide a comprehensive picture of the data assets reported by the companies? A data asset is an asset that is formed from data elements by adding value, not just a collection of data.Please elaborate.

3.How do companies with sample databases determine that their assets include data assets? What percentage of data assets?Please elaborate.

4.In assessing the valuation of data assets of listed companies in China from 2015-2023, the final valuation results were not seen using the data asset valuation model created based on the productive artificial intelligence model created in this thesis. Please add.

Reviewer #2: The article highlights the dynamic nature of data assets and the need for more adaptive valuation models in the digital economy.

The proposed framework (integrating data feature extraction, value generation algorithms, and market adaptability assessment), offers a comprehensive solution for more precise and responsive valuation. Additionally, the discussion on how Generative AI enhances data valuation, through synthetic data generation, improved model training, and market adaptability, provides valuable insights into its broader implications. The introduction of a systematic framework for dynamic pricing, along with its potential applications for businesses and policymakers, makes this study a relevant contribution to the field of digital transformation and data asset management.

This article offers important perspectives on leveraging data as a strategic business asset.

However, the empirical data collection process is not clearly explained, and the data itself is not yet accessible. It is only mentioned "The sample primarily originated from Chinese

165 A-share listed companies, covering the period from 2015 to 2023." (line 164-165) and "The calculation of data asset value was based on corporate financial statements and industry analysis reports" (Line 182). A more detailed explanation of the data collection process is necessary to ensure transparency and enhance the credibility of the study. While the article references Chinese A-share listed companies as the dataset, it does not specify the exact sources of this data and other listed variable.

It would be beneficial to specify the exact source of the A-share listed companies used in the study. Providing this information, along with access to the data, would offer greater transparency and allow for a more thorough evaluation of the model’s robustness. Additionally, making the data available would enhance the study’s replicability and reproducibility, further strengthening its impact. Clarifying the origin of the dataset would help readers assess the reliability and scope of the data used in model validation.

**Do you want your identity to be public for this peer review?** For information about this choice, including consent withdrawal, please see our Privacy Policy

Reviewer #1: No

Reviewer #2: No

---

## [Author Response · Author response to Decision Letter 1]

25 Apr 2025

Response to Reviewers

We responded to each issue raised by the reviewers in a bold and underlined manner.

Reviewers' comments:

1. Is the manuscript technically sound, and does the data support the conclusions?

Reviewer #1: Partly

Reviewer #2: Yes

Response: For Reviewer 1’s response of “Partly”, we summarized what has been conducted. The data asset valuation model is constructed using a Generative AI algorithm based on real data. To construct a robust and practical model, widely adopted machine learning models are used to pick influential predictive variables. The constructed model is tested using validation and test data sets. Also, it is shown that fusion models are more capable than single models.

2. Has the statistical analysis been performed appropriately and rigorously?

Reviewer #1: Yes

Reviewer #2: Yes

Response: No responses are required

3. Have the authors made all data underlying the findings in their manuscript fully available?

The PLOS Data policy requires authors to make all data underlying the findings described in their manuscript fully available without restriction, with rare exceptions (please refer to the Data Availability Statement in the manuscript PDF file). The data should be provided as part of the manuscript or its supporting information, or deposited to a public repository. For example, in addition to summary statistics, the data points behind means, medians, and variance measures should be available. If there are restrictions on publicly sharing data—e.g. participant privacy or use of data from a third party—those must be specified.

Reviewer #1: Yes

Reviewer #2: No

Response: For Reviewer 2’s response of “No”, we provide the link for the data set: https://www.hkex.com.hk/Mutual-Market/Stock-Connect/Eligible-Stocks/A-share-Lookup-Tools?sc_lang=en. However, we do not share the collected and cleansed data set in the article, instead the data set was shared in China macroeconomic database.

4. Is the manuscript presented in an intelligible fashion and written in standard English?

Reviewer #1: Yes

Reviewer #2: Yes

Response: No responses are required.

5. Review Comments to the Author

Reviewer #1: This article builds a generative artificial intelligence-based data asset valuation model based on existing literature, which integrates data feature extraction, a generative value generation algorithm, and market adaptive valuation to dynamically assess the business value of data assets. The conclusions of the article show that the model can address the shortcomings of traditional valuation methods in data asset valuation. Overall, the article is well written, the analysis is solid and the conclusions are challenging and bring a lot of new information to the field.

However, there are certain issues, such as the sample and analysis methods of the study, that deserve in-depth discussion. There are four issues that require further research.

1. What is the basis for the composition of the three dimensions of the model? What are the criteria for dimension selection? Please elaborate. Please elaborate.

Response: All three dimensions are critically significant in the construction data asset valuation model. Each has a crucial role in refining the contracted data asset valuation model. We do not use any criteria in dimension selection. However, the related literature gives us insight into the dimensions of the model.

2. Do metrics such as user data, market data, and operational data provide a comprehensive picture of the data assets reported by the companies? A data asset is an asset that is formed from data elements by adding value, not just a collection of data. Please elaborate.

Response: Yes, user data, market data, and operational data provide a picture for data assets, however, their role is complimentary. Their collection process and accessibility to them are also a part of the whole process.

3. How do companies with sample databases determine that their assets include data assets? What percentage of data assets? Please elaborate.

Response: Actually, companies keep these records for some regularity or obligatory conditions with no attention to their value of them. However, the currently held databases are a part of the data assets. How much portion of the data asset is not computed for the sample data?

4. In assessing the valuation of data assets of listed companies in China from 2015-2023, the final valuation results were not seen using the data asset valuation model created based on the productive artificial intelligence model created in this thesis. Please add.

Response: Significant variables are found that play a critical role in the data asset valuation model. Return on Assets (ROA), Return on Equity (ROE), and Operating Cash Flow (OCF) are found. Besides, in the conclusion section, these findings are stated in 2) verbally.

Reviewer #2: The article highlights the dynamic nature of data assets and the need for more adaptive valuation models in the digital economy.

The proposed framework (integrating data feature extraction, value generation algorithms, and market adaptability assessment), offers a comprehensive solution for more precise and responsive valuation. Additionally, the discussion on how Generative AI enhances data valuation, through synthetic data generation, improved model training, and market adaptability, provides valuable insights into its broader implications. The introduction of a systematic framework for dynamic pricing, along with its potential applications for businesses and policymakers, makes this study a relevant contribution to the field of digital transformation and data asset management.

This article offers important perspectives on leveraging data as a strategic business asset. However, the empirical data collection process is not clearly explained, and the data itself is not yet accessible. It is only mentioned that "The sample primarily originated from Chinese 165 A-share listed companies, covering the period from 2015 to 2023" (lines 164-165) and "The calculation of data asset value was based on corporate financial statements and industry analysis reports" (Line 182).

A more detailed explanation of the data collection process is necessary to ensure transparency and enhance the credibility of the study. While the article references Chinese A-share listed companies as the dataset, it does not specify the exact sources of this data and other listed variables.

It would be beneficial to specify the exact source of the A-share listed companies used in the study. Providing this information, along with access to the data, would offer greater transparency and allow for a more thorough evaluation of the model’s robustness.

Additionally, making the data available would enhance the study’s replicability and reproducibility, further strengthening its impact. Clarifying the origin of the dataset would help readers assess the reliability and scope of the data used in model validation.

Response: We provide the link for the data set: https://www.hkex.com.hk/Mutual-Market/Stock-Connect/Eligible-Stocks/A-share-Lookup-Tools?sc_lang=en. However, we do not share the collected and cleansed data set in the article, instead the data set was shared in China macroeconomic database.

6. PLOS authors have the option to publish the peer-review history of their article (what does this mean?). If published, this will include your full peer review and any attached files.

If you choose “no”, your identity will remain anonymous, but your review may still be made public.

Do you want your identity to be public for this peer review? For information about this choice, including consent withdrawal, please see our Privacy Policy.

Reviewer #1: No

Reviewer #2: No

Response: No responses are required.

---

## [Decision Letter · Decision Letter 1]

12 Jun 2025

Dear Dr. Tang,

Thank you for submitting your manuscript to PLOS ONE. After careful consideration, we feel that it has merit but does not fully meet PLOS ONE’s publication criteria as it currently stands. Therefore, we invite you to submit a revised version of the manuscript that addresses the points raised during the review process.

We look forward to receiving your revised manuscript.

Kind regards,

Fu-Sheng Tsai

Academic Editor

PLOS ONE

Reviewers' comments:

Reviewer's Responses to Questions

**Comments to the Author**

Reviewer #1: All comments have been addressed

Reviewer #2: All comments have been addressed

2. Is the manuscript technically sound, and do the data support the conclusions?

Reviewer #1: Yes

Reviewer #2: Yes

3. Has the statistical analysis been performed appropriately and rigorously?

Reviewer #1: Yes

Reviewer #2: Yes

4. Have the authors made all data underlying the findings in their manuscript fully available?

Reviewer #1: Yes

Reviewer #2: No

5. Is the manuscript presented in an intelligible fashion and written in standard English?

Reviewer #1: Yes

Reviewer #2: Yes

Reviewer #1: (No Response)

Reviewer #2: My recommendation is minor: the manuscript should include detailed information about the data source and the specific steps taken in data collection and preparation, including any filters applied and the data cleansing process. While the cleaned dataset itself does not need to be shared, a clear explanation of the procedures followed is necessary.

In line 179, the authors mention that the data are acquired from Chinese A-share listed companies. However, the explanation is very minimal. Since the data is publicly available and the source has been stated in the response letter (https://www.hkex.com.hk/Mutual-Market/Stock-Connect/Eligible-Stocks/A-share-Lookup-Tools?sc_lang=en), I recommend that the authors also include this citation in the manuscript. This addition is important for clarity, transparency, and reproducibility of the study.

**Do you want your identity to be public for this peer review?** For information about this choice, including consent withdrawal, please see our Privacy Policy

Reviewer #1: No

Reviewer #2: **Yes: ** Muhamad Prabu Wibowo

---

## [Author Response · Author response to Decision Letter 2]

18 Jun 2025

Response to Reviewers

Reviewers' comments:

Reviewer's Responses to Questions

Comments to the Author

1. If the authors have adequately addressed your comments raised in a previous round of review and you feel that this manuscript is now acceptable for publication, you may indicate that here to bypass the “Comments to the Author” section, enter your conflict of interest statement in the “Confidential to Editor” section, and submit your "Accept" recommendation.

Reviewer #1: All comments have been addressed

Reviewer #2: All comments have been addressed

2. Is the manuscript technically sound, and do the data support the conclusions?

Reviewer #1: Yes

Reviewer #2: Yes

3. Has the statistical analysis been performed appropriately and rigorously?

Reviewer #1: Yes

Reviewer #2: Yes

4. Have the authors made all data underlying the findings in their manuscript fully available?

The PLOS Data policy requires authors to make all data underlying the findings described in their manuscript fully available without restriction, with rare exceptions (please refer to the Data Availability Statement in the manuscript PDF file). The data should be provided as part of the manuscript or its supporting information, or deposited in a public repository. For example, in addition to summary statistics, the data points behind means, medians, and variance measures should be available. If there are restrictions on publicly sharing data—e.g. participant privacy or use of data from a third party—those must be specified.

Reviewer #1: Yes

Reviewer #2: No

Response to Reviewer #2: We provide the link for the implemented dataset in the manuscript. Also, we provide a brief explanation for how the total dataset is reduced to the final form to be used in the modeling process. However, we do not share the preprocessed dataset. If requested by readers via email to the corresponding author, we can email it in unprocessed form, since preprocessing is an important part of the modeling process and a problem-specific aspect.

5. Is the manuscript presented in an intelligible fashion and written in standard English?

Reviewer #1: Yes

Reviewer #2: Yes

6. Review Comments to the Author

Reviewer #1: (No Response)

Reviewer #2: My recommendation is minor: the manuscript should include detailed information about the data source and the specific steps taken in data collection and preparation, including any filters applied and the data cleansing process. While the cleaned dataset itself does not need to be shared, a clear explanation of the procedures followed is necessary.

In line 179, the authors mention that the data are acquired from Chinese A-share listed companies. However, the explanation is very minimal. Since the data is publicly available and the source has been stated in the response letter (https://www.hkex.com.hk/Mutual-Market/Stock-Connect/Eligible-Stocks/A-share-Lookup-Tools?sc_lang=en), I recommend that the authors also include this citation in the manuscript. This addition is important for clarity, transparency, and reproducibility of the study.

Response: Lines 163-167 include how the dataset is filtered out. Since we apply strict rules to eliminate problematic data, for instance, missing observations and outliers are out completely, the final dataset obtained is composed of directly usable data in the modeling process. The second sentence is also added to the text.

7. PLOS authors have the option to publish the peer review history of their article (what does this mean?). If published, this will include your full peer review and any attached files.

Do you want your identity to be public for this peer review? For information about this choice, including consent withdrawal, please see our Privacy Policy.

Reviewer #1: No

Reviewer #2: Yes: Muhamad Prabu Wibowo

---

## [Decision Letter · Decision Letter 2]

9 Jul 2025

Data Asset Valuation Model Based on Generative Artificial Intelligence

PONE-D-24-51382R2

Dear Dr. Tang,

We’re pleased to inform you that your manuscript has been judged scientifically suitable for publication and will be formally accepted for publication once it meets all outstanding technical requirements.

Kind regards,

Fu-Sheng Tsai

Academic Editor

PLOS ONE

Additional Editor Comments (optional):

Reviewers' comments:

Reviewer's Responses to Questions

**Comments to the Author**

Reviewer #1: All comments have been addressed

Reviewer #2: All comments have been addressed

2. Is the manuscript technically sound, and do the data support the conclusions?

Reviewer #1: Yes

Reviewer #2: Yes

3. Has the statistical analysis been performed appropriately and rigorously?

Reviewer #1: Yes

Reviewer #2: Yes

4. Have the authors made all data underlying the findings in their manuscript fully available?

Reviewer #1: Yes

Reviewer #2: Yes

5. Is the manuscript presented in an intelligible fashion and written in standard English?

Reviewer #1: Yes

Reviewer #2: Yes

Reviewer #1: (No Response)

Reviewer #2: (No Response)

**Do you want your identity to be public for this peer review?** For information about this choice, including consent withdrawal, please see our Privacy Policy

Reviewer #1: No

Reviewer #2: **Yes: ** Muhamad Prabu Wibowo

---

## [Editor Report · Acceptance letter]

PONE-D-24-51382R2

PLOS ONE

Dear Dr. Tang,

I'm pleased to inform you that your manuscript has been deemed suitable for publication in PLOS ONE. Congratulations! Your manuscript is now being handed over to our production team.

Kind regards,

on behalf of

Professor Fu-Sheng Tsai

Academic Editor

PLOS ONE